# The Fundamental Tension in Integrated Information Theory 4.0’s Realist Idealism

**DOI:** 10.3390/e25101453

**Published:** 2023-10-16

**Authors:** Ignacio Cea, Niccolo Negro, Camilo Miguel Signorelli

**Affiliations:** 1Center for Research, Innovation and Creation, Temuco Catholic University, Temuco 4813302, Chile; 2Faculty of Religious Sciences and Philosophy, Temuco Catholic University, Temuco 4813302, Chile; 3School of Psychological Sciences, Tel Aviv University, P.O. Box 39040, Tel Aviv 6997801, Israel; niccolo.negro.research@gmail.com; 4FNRS, GIGA Institute, University of Liege, Rue d’Egmont 5, B-1000 Brussels, Belgium; camilo.miguel.signorelli@gmail.com; 5Department of Computer Science, University of Oxford, Wolfson Building, Parks Road, Oxford OX1 3QD, UK

**Keywords:** consciousness, realism, idealism, integrated information theory, ontology of consciousness, scientific metaphysics

## Abstract

Integrated Information Theory (IIT) is currently one of the most influential scientific theories of consciousness. Here, we focus specifically on a metaphysical aspect of the theory’s most recent version (IIT 4.0), what we may call its *idealistic ontology*, and its tension with a kind of *realism* about the external world that IIT also endorses. IIT 4.0 openly rejects the mainstream view that consciousness is generated by the brain, positing instead that consciousness is ontologically primary while the physical domain is just “operational”. However, this philosophical position is presently underdeveloped and is not rigorously formulated in IIT, potentially leading to many misinterpretations and undermining its overall explanatory power. In the present paper we aim to address this issue. We argue that IIT’s idealistic ontology should be understood as a specific combination of phenomenal primitivism, reductionism regarding *Φ*-structures and complexes, and eliminativism about non-conscious physical entities. Having clarified this, we then focus on the problematic tension between IIT’s idealistic ontology and its simultaneous endorsement of realism, according to which there is some kind of external reality independent of our minds. After refuting three potential solutions to this theoretical tension, we propose the most plausible alternative: understanding IIT’s realism as an assertion of the existence of other experiences beyond one’s own, what we call a *non-solipsistic idealist realism*. We end with concluding remarks and future research avenues.

## 1. Introduction

Currently, Integrated Information Theory (IIT) is one of the leading scientific theories of consciousness [1,2,3,4]. IIT is especially interesting because it aims to explain the phenomenal, subjective character of experience; not its behavioural, computational or functional correlates [5]. However, the consistency of the theory has been critically addressed on several occasions [6,7,8,9,10]. Here, we focus specifically on a metaphysical aspect of the theory’s most recent version (IIT 4.0) [11,12,13,14], what we may call its *idealistic ontology*, and its tension with a kind of *realism* about the external world that IIT also endorses. IIT 4.0 openly rejects the mainstream view that consciousness is generated by the brain, positing instead that consciousness is ontologically primary while the physical domain is just an experience-dependent “operational” construct to understand consciousness from the scientific, third-person perspective:

“The primacy of intrinsic existence (of experience) in IIT contrasts with standard attempts at accounting for consciousness as something ‘generated by’ or ‘emerging from’ a substrate constituted of matter and energy and following physical laws.” [12] (p. 38)

However, this philosophical position is currently underdeveloped and is not rigorously formulated in IIT, potentially leading to many misinterpretations and undermining its overall explanatory power. In the present paper we aim to address this issue. After presenting the basic building blocks of the theory (Section 2), in Section 3 we argue that IIT’s idealistic ontology should be understood as a specific combination of: (i) *phenomenal primitivism* (i.e., consciousness is ontologically primordial, irreducible to and non-derivable from anything else); (ii) *reductionism* regarding *Φ*-structures and complexes (i.e., the material substrate and its unfolded causal structure ontologically reduce to consciousness); and (iii) *eliminativism* about non-conscious physical entities such as, presumably, atoms, neurons, bodies and rocks (i.e., all physical entities that are not substrates of consciousness do not truly exist). Then, in Section 4, we focus on the problematic tension between IIT’s idealistic ontology and its simultaneous endorsement of *realism,* namely, the idea that “something exists (and persists) independently of our own experience” [12] (p. 6). Prima facie, it seems a contradiction to hold both that all that exist are ultimately conscious experiences, and that there are things existing independently of our consciousnesses. After presenting and refuting three potential solutions to this theoretical tension, we propose what we regard as the most plausible alternative: understanding IIT’s realism as an assertion of the existence of other experiences beyond one’s own, what we call a *non-solipsistic idealist realism.* We also evaluate the resulting metaphysical picture in light of Chalmers’ distinction between *realist* and *anti-realist idealisms* [15], and concordantly conclude that it qualifies as a version of the former. We end with concluding remarks and future avenues of research.

## 2. IIT Basics

IIT starts from a purportedly self-evident characterization of the essential properties of conscious experience, directly accessible from the first-person perspective: the “*axioms of phenomenal existence*” (or “*phenomenal axioms*”) [11,12]. Its 0th axiom is that *consciousness exists;* it is immediately known to exist, beyond any doubt. The existence of everything else is inferred from this “fundamental and certain starting point” [11] (p. 3). Then, IIT adds the following five phenomenal axioms: consciousness (i) exists *intrinsically*: for itself, as inherently subjective; (ii) is *informative* or specific; (iii) is *integrated* (i.e., a unitary whole irreducible to its parts); (iv) is *exclusive* or definite; and (v) is *composed*: any experience is a structure of phenomenal distinctions (e.g., colours, shapes, thoughts, affects, etc.) and phenomenal relations (e.g., the rounded shape is white, the alarm is noisy and displeasing, etc.)

From this characterization of consciousness, IIT derives its main theoretical postulates, the “postulates of physical existence” [12] (p. 4). These postulates are meant to translate the phenomenal axioms into operationalized properties that are amenable to scientific manipulation and observation, and which are claimed to be individually necessary and jointly sufficient for any candidate physical system to be conscious; that is, to qualify as the *physical substrate of consciousness* (hereafter “PSC”, also called *complex* or *maximal substrate*).

Although previous versions of IIT did not explicitly clarify the logic of this translation (for discussions on this point, see [16,17,18,19]), IIT 4.0 explains that the postulates are inferred via an “inference to a good explanation” from the axioms and a triad of basic assumptions [11,12]. These assumptions are *realism, operational physicalism,* and *atomism*. According to realism, there is a mind-independent reality; according to operational physicalism, phenomenal properties must be operationalized in terms of “cause–effect power”, which is assessed through manipulations and observations that conscious observers can do on candidate physical substrates; according to atomism, manipulations and interventions should be carried over the minimal building blocks that can be manipulated.

The 0th postulate of physical existence, which mirrors the 0th axiom, is the “‘principle of being’: to exist physically means to have *cause–effect power*—being able to take and make a difference” [11] (p. 3). Then, for each one of the other five axioms there is a corresponding postulate in which each of the essential properties of phenomenal existence (e.g., intrinsic existence, information, etc.) is translated into a physical property of cause–effect power (e.g., intrinsic and specific cause–effect power, etc.). These physical postulates are summarized by the claim that any conscious physical system, such as, for example, a neural network in the posterior areas of the brain, must be “a maximum of irreducible, specific, compositional, intrinsic cause–effect power” [20] (p. 3).

Based on the mathematical formalization of these postulates, IIT provides the tools to measure the extent to which a physical system exists intrinsically (and irreducibly, specifically, etc.). This is computed as the value *φ*_s_ (amount of “System Integrated Information”, introduced in [13]). Then, following IIT’s *principle of maximal existence* (i.e., “what exists is what exists the most”), among overlapping candidate systems specifying *φ*_s_ > 0, the substrate that qualifies as the PSC and purportedly enjoys intrinsic, absolute, phenomenal existence, is the one that specifies the largest value of *φ*_s_ (symbolized by *φ*_s_*).

Once the PSC is identified, the particular structure of its intrinsic causal powers is unfolded into a *Φ-structure* (or *cause–effect structure*)*,* which is constituted by all of the maximally irreducible, specific, intrinsic causal powers of the mechanisms of the system (i.e., the “causal distinctions”) and the relations they bear to each other (i.e., the “causal relations”) [12,14,21,22]. Each causal distinction and causal relation specified by the PSC in its current state has an associated value of integrated information *φ*_d_ (for distinctions) and *φ*_r_ (for relations). Then, according to IIT, the sum total of these *φ* values quantifies the *structured information* (*Φ*) of the *Φ*-structure specified by the PSC [12].

Crucially, IIT claims that the *Φ*-structure and its *Φ* value account for the content and level of consciousness respectively: “If a system S in state* s is a complex [i.e., a PSC], then its *Φ*-structure corresponds to the quality of the experience of S in state s, while its *Φ* value corresponds to its quantity” [12] (p. 28). This expresses IIT’s ‘central identity’ between consciousness and integrated information; more specifically, between phenomenal experience and the *Φ*−structure of the PSC [11,12,20,23]. Intriguingly, and in contrast to previous versions of the theory, the identity is now explicitly emphasized as an *explanatory* identity [5,11,12,22]. This means that all of the phenomenal properties of consciousness “have a good explanation in terms of the specific physical properties of the corresponding cause–effect structure” [12] (p. 2). In this way, IIT provides a mathematical formalism and experimental methodology to operationalize experience in physical terms, such that, thanks to the explanatory identity, “the intrinsic (subjective) feeling of the experience can be explained extrinsically (objectively, i.e., operationally or physically) in terms of cause–effect power” [12] (p. 6).

## 3. IIT’s Idealistic Ontology: Reducing and Eliminating the Physical

In the present section, we are going to address the ontological dimension of IIT. Recall that IIT 4.0 explicitly rejects the idea that consciousness is generated by the brain, positing instead that consciousness is ontologically primary while the physical domain is just “operational”. Our analysis will suggest that IIT 4.0 not only takes consciousness to be ontologically primitive (Section 3.1) but eliminates non-conscious physical entities (Section 3.2), reduces *Φ*-structures to their corresponding experiences (Section 3.3), and also reduces physical substrates of consciousness (PSCs) to their *Φ*-structures, and therefore, ultimately, to experiences as well (Section 3.4).

Before presenting in detail our analysis of IIT’s ontology, it is important to clarify, from the outset, the distinction between ontological reduction and elimination. ‘X reduces to Y’ means that instead of X and Y being two distinct, objectively existing entities, there is just one entity, and both terms “X” and “Y” refer to that entity in reality (e.g., instead of there being genes and DNA molecules, our ontology is reduced: there are only DNA molecules, but the terms “genes” and “DNA molecules” can both be used to refer to them in different contexts). Elimination, instead, means that a term that was thought to refer to something real existing out there, does not in fact refer, it points to nothing in reality (e.g., “phlogiston”, a hypothetical substance that purportedly explained combustion, was not subsequently reduced to anything; it was ontologically eliminated) [24,25].

### 3.1. Phenomenal Existence Is True Existence: Experience as Ontologically Primitive

We start addressing the ontological status of consciousness according to IIT. Recall that the 0th and 1st phenomenal axioms state that consciousness exists intrinsically, for itself, and that this phenomenal existence is the only type of existence that is given immediately, beyond any doubt. Now, if consciousness exists intrinsically, and in a way that is self-evident and immediately known, then, according to IIT adherents, consciousness is “the fundamental, the ultimate” [19] (p. 21), something “ontologically basic, in the sense that it exists fundamentally” [26] (p. 5).

So, IIT asserts that consciousness has a radical ontological primacy, but there is more: only experiences “are what truly exist” [11] (p. 16). We will call this idea IIT’s *principle of true existence* (or “PTE”): only phenomenal existence is true existence; consciousness is “the only existence worth having—what we might call true existence… [experience] truly exists because it exists for itself—it exists absolutely” [11] (pp. 8–9). In [12], this idea is expressed when Albantakis and colleagues assert that “what truly exists is… an intrinsic entity [i.e., an experience] that exists for itself, absolutely, rather than relative to an external observer” (p. 28). In contrast to other metaphysical principles (e.g., principle of being, maximal existence, etc.), what we call the principle of true existence (PTE) is not explicitly formulated as a principle in IIT, but is implicitly assumed in the 4.0 version, playing a huge role in IIT’s account of free will, according to which consciousness is the only true cause because it is the only truly existing entity [11] (more on this in Section 3.2 below). Moreover, this has already been suggested in the first version of the theory:

“We are by now used to considering the universe as a vast empty space that contains enormous conglomerations of mass, charge, and energy…However… an equally valid view of the universe is this: a vast empty space that contains mostly nothing, and occasionally just specks of integrated information (*Φ*) [i.e., consciousness]... In fact, it may be more valid, since to be highly conscious (to have high *Φ*) implies that there is something it is like to be you… From this standpoint, it would seem that entities with high *Φ* exist in a stronger sense than entities of high mass” [27] (p. 233)

In other words, IIT’s long-held intuition underlying the PTE is that the level of consciousness of a system is a better measure of its existence than its conventional physical properties because only the former entails that the system “exists for itself”. This intuition has only grown stronger in IIT, to such an extent that, quoting Schrodinger, IIT proponents write that a world with no consciousness “would be ‘a play before empty benches, not existing for anybody, thus quite properly speaking not existing’” [11] (p. 8).

More technically, IIT 4.0 now distinguishes between intrinsic and extrinsic entities. The former are conscious entities, operationalized as *Φ*-structures with an associated value of structured information *Φ*, specified by systems with maximal system integrated information (*φ*_s_*). Extrinsic entities, in contrast, are physical things that do not specify maximal system integrated information, such as bodies, neurons and chairs, all of which are not conscious. Intrinsic entities have intrinsic existence (i.e., exist in an absolute sense, for themselves, indubitably, truly), while extrinsic entities have extrinsic existence (i.e., exist in a weaker, relative sense, not for themselves, but for external observers) [11,12,28]. Hence, consciousness and intrinsic existence coincide: consciousness is ontologically fundamental or primordial insofar as it is the only thing that exists for itself, absolutely, truly; while everything else, presumably including electrons and neurons, only exist in a weaker, relative sense:

“Between intrinsic and extrinsic existence, then, passes the most fundamental of divides—the great divide of being. This is the divide between what truly exists in an absolute sense, in and of itself—namely conscious, intrinsic entities—and what only exists in a relative sense, for something else.” [11] (p. 8)

IIT is therefore not “just” a theory of consciousness, it is also a theory of existence, so its ontology must be analysed. In order to clarify its ontological implications, we now examine the weaker, relative sense of existence reserved for non-conscious entities.

### 3.2. The Ontological Elimination of Non-Conscious Physical Entities

We turn to the question: what does IIT’s absolute/relative distinction regarding existence really amount to? Is it plausible to hold that both are genuine forms of existence? A first possible reading is a Meinongian one, according to which extrinsic entities belong to the realm of being, but do not have the property of existing [29]. However, Albantakis and colleagues [12] (p. 41) state explicitly that existence is not a property, so this is not a viable option. A more promising interpretation of IIT’s ontology is to conclude that only conscious, intrinsic existence is real existence, while the extrinsic “existence” of systems that do not specify maximal *φ*_s_ is in fact a kind of unreal or untrue existence, that is, non-existence. Recall that only intrinsic, phenomenal existence is “true existence” [11] (p. 8), what we called IIT’s principle of true existence (Section 3.1). By implication, then, extrinsic, relative existence would not be true existence; non-conscious physical entities would not really exist according to IIT, not if realism regarding the existence of an entity requires its mind-independent existence [30]. In other words, physical systems that are not substrates of consciousness only “exist”, at best, in the experience of an observer that assesses their causal power through causal manipulations and observations. They only exist relatively, untruly, and hence, do not really exist: “Bodies and organs, tables and rocks, stars and planets… are likely to unfold into extrinsic entities… They only exist vicariously, from the perspective of some intrinsic entity, and so *they do not truly exist*” [11] (p. 8, italics added).

As radical as it may be, from the ontological perspective of IIT, all non-conscious physical entities like bodies and planets are like the phlogiston, or the ether of ancient cosmologies; things we can conceptualize, imagine, seemingly perceive, and that may even play some explanatory roles in scientific theorizing, but which do not actually exist beyond our minds. Interestingly, this “eliminative idealism” is a radical metaphysical inversion of the well-known eliminative materialism of the Churchlands [31,32], as well as of illusionism [33,34,35]. IIT claims, unlike these two, that what is ontologically eliminated or illusory are ultimately non-conscious physical entities such as brain scanners, unconscious parts of brains, and bodies, instead of consciousness.

This radical eliminative idealistic reading is additionally supported by the following considerations. First, it is the most plausible way to make sense of IIT’s account of free will [11]. The latter is based on the claim that conscious decisions and intentions are not causally excluded by the underlying neural processes, and hence not rendered epiphenomenal, because conscious decisions and intentions do not compete, causally speaking, with neural processes in determining the actions of an agent. Why? because only consciousness really exists: “as a conscious being, I truly exist and truly cause, whereas my neurons or my atoms neither truly exist nor truly cause” [11] (p. 2).

Second, it is strongly suggested by IIT’s assumption that physicalism should be seen as an “operational” view, according to which physical terms such as “neurons” and “electrons” are good descriptive and explanatory instruments to produce scientific knowledge about, in this case, consciousness, but they do not refer to any truly existing thing in reality: “unlike phenomenal existence… physical existence is an explanatory construct (a postulate) and it is assessed operationally from within consciousness” [12] (p. 2). The same is suggested by Chis-Ciure [19]: “when it comes to ontology, consciousness has primacy … whereas cause-effect power has an explanatory nature (is instrumental)” (p. 11).

In philosophy of science terms, IIT’s operational physicalism would indicate that the theory endorses a form of instrumentalism, specifically regarding physical entities. Generally speaking, instrumentalism claims that scientific theories are fundamentally, powerful cognitive tools to predict, control and explain phenomena within a certain domain of inquiry, but not true or false descriptions of reality. The latter makes instrumentalism a form of anti-realism, most commonly about non-observable theoretical entities [36,37] (more on this in Section 4.1). Following this, IIT’s instrumentalism about the physical realm would mean that it considers physical entities as useful constructs to advance the science of consciousness, without committing to either the truth or falsity of their objective reality. However, when combined with the principle of true existence (i.e., only phenomenal existence is true existence), and IIT’s view that only a specific class of physical entities enjoy phenomenal existence (i.e., those specifying maximal *φ*_s_), it follows that IIT considers non-conscious physical entities just as useful explanatory instruments, and not as truly existing entities.

In sum, IIT’s endorsement of the ontological primacy of experience comes also with the ontological elimination of every non-conscious entity, such as, presumably, atoms, neurons, bodies and chairs, because they do not truly exist, given that they do not exist consciously, for themselves, as indicated by their non-maximal *φ*_s_ values. This is also supported by IIT’s recent account of free will and its operational physicalism.

Having clarified this point, we turn now to the evaluation of the ontological status of conscious physical entities: *Φ*-structures (Section 3.3) and Physical Substrates of Consciousness (PSCs, Section 3.4).

### 3.3. The Ontological Reduction of Φ-Structures to Subjective Experiences

If our analysis of IIT 4.0 is correct, the theory is *eliminative* with respect to conventional physical entities that do not specify maximal system integrated information and hence do not qualify as conscious, but *reductive* with respect to the *sui generis* type of physical entities called *Φ*-structures. As mentioned in Section 2, *Φ*-structures are unfolded causal structures constituted by interrelated cause-effect powers that the mechanisms of a physical substrate exert intrinsically, i.e., within the system, in a specific, structured, and maximally irreducible way [11,12,20,23]. Also, notice that given that the physical is defined precisely in terms of causal power (IIT’s principle of being), *Φ*-structures qualify as physical entities [11,12,26].

Crucially, IIT’s algorithmic procedure is designed to reveal these structures and their degree of irreducibility (structured information: *Φ*), which otherwise would remain veiled to a third-person perspective, precisely because they are the way in which the physical substrate of consciousness (e.g., a conscious neural network) exists for itself, intrinsically, and hence truly: “what exists here and now is the *Φ*-structure… [it] is what exists in physical terms… [it] truly exists because it exists for itself—it exists absolutely as a conscious being” [11] (pp. 7–9). But this statement that the *Φ*-structure truly exists because it exists for itself suggests that a system’s subjective experience and its *Φ*-structure are one and the same thing.

Evidently, this requires casting a deeper look at IIT’s ‘central identity’ between consciousness and *Φ*-structure, and understanding what metaphysical commitments come with it. As mentioned in Section 2, in recent years IIT has been emphasizing that this identity should be understood specifically as an *explanatory* identity [5,11,12,22]. Now, although there might be different ways to interpret the metaphysical relation underlying this explanatory identity, we think that the identity must be considered both as explanatory and metaphysical, that is, as a *metaphysical identity with explanatory power*.

Without a metaphysically charged notion of identity, IIT might face the following scenario. In every conscious physical system, two distinct entities with intrinsic existence—the system’s experience and its *Φ*-structure—coexist, overlapped over the same space and time, i.e., a sort of unpalatable ontological dualism regarding the experience/*Φ*-structure relation. Moreover, both entities would be distinct, but equally true forms in which a PSC exists for itself. We take these implications to be both unparsimonious and confusing.

A far more elegant and clearer alternative is to assert that the terms “conscious experience” and “*Φ*-structure” are two distinct ways to describe how a conscious entity exists for itself: *phenomenologically* in the case of the former, as grasped from the first-person point of view; *physically* in the case of the latter, from the third-person perspective. This reading would also be the most straightforward interpretation of IIT’s claims that “the physical correspondent of an experience is not the substrate as such but the *Φ*-structure specified by the substrate” [12] (p. 38); that “what exists here and now is the *Φ*-structure… it exists absolutely as a conscious being” [11] (pp. 7–9); or that “being an intrinsic entity, properly defined [as a *Φ*-structure], is one and the same thing as being conscious” [28] (p. 623). 

Nonetheless, there are some rare, but explicit, rejections of the metaphysical nature of the identity that have been offered on the grounds that consciousness is ontologically ultimate and hence cannot be reduced to a physical entity such as a *Φ*-structure: “the identity… should be understood as explanatory *rather than* metaphysical” [38] (p. 52, italics modified, see also [39]). We think that this worry is unproblematic and a symptom of previous difficulties in admitting the full-blown idealistic implications of the theory: experiences and *Φ*-structures are metaphysically one and the same, but that does not entail that consciousness reduces to physical structures of intrinsic causal powers; on the contrary, conscious experiences are primary, and cause–effect structures reduce to them! In other words, there is no *Φ*-structure in addition to its associated experience, but both the terms “consciousness” and “*Φ*-structure” refer to the same one thing in reality. However, the former term is the one that most directly captures the essence of the phenomenon as revealed subjectively. “*Φ*-structure”, in contrast, is the operational, indirect, but scientifically useful way of referring to subjective experience from the third-person perspective.

Therefore, even if strictly speaking any identity is symmetrical, we think that an appropriate interpretation of IIT takes *Φ*-structures as being ontologically reduced to experiences, rather than vice-versa. This halves the number of entities that are posited by an alternative, dualist interpretation of IIT’s explanatory identity, which takes *Φ*-structures and experiences as two distinct entities. Instead, for the monist interpretation we favour, which gives metaphysical import to IIT’s central identity, only experiences exist, but can also be conveniently described and (purportedly) explained in scientific terms as the unfolded *Φ*-structures of the corresponding PSCs. The question we turn to now is about the ontological status of these PSCs.

### 3.4. The Ontological Reduction of Physical Substrates of Consciousness (PSCs) to Subjective Experiences

As clearly stated in IIT 4.0, a physical system that specifies a positive amount of system integrated information *φ*_s_ (i.e., that has causal power over itself as a unitary irreducible whole), enjoys “irreducible existence” [12] (p. 18). However, given IIT’s exclusion postulate and principle of maximal existence, for overlapping networks with positive system-integrated information, the one that “really” exists is the one that “exists the most”. That is, the system that exists is the one “with the maximum value of system integrated information… [while the others are] excluded from existence” [12] (p. 18). Hence, according to IIT, there is a class of physical systems that really exist irreducibly: networks that specify maximum system integrated information (*φ*_s_*). These systems qualify as physical substrates of consciousness (PSCs) [12].

However, PSCs, while being maximally mereologically irreducible, i.e., existing as unitary wholes maximally irreducible to their parts, are not irreducible to their *Φ*-structures: “what actually exists is only the *Φ*-structure corresponding to my experience, not also an associated physical substrate” [11] (p. 10). More specifically, PSCs exist intrinsically as *Φ*-structures: “a complex [i.e., a PSC] does not exist as such but only “unfolded” as a *Φ*-structure—an *intrinsic entity* that exists for itself, absolutely” [12] (p. 37); “the substrate does not exist as such, separately from the cause–effect structure it specifies; rather, it exists *as* that structure” [28] (p. 631).

While speaking in terms of “existing as” may be subject to diverse interpretations, arguably the most straightforward is that PSCs do not exist as entities in their own right, in addition to their *Φ*-structures. PSCs are, intrinsically, nothing but *Φ*-structures. This is reinforced by IIT’s statement that

“a substrate is what can be observed and manipulated “operationally” from the extrinsic perspective. From the intrinsic perspective, what truly exists is a complex with all its causal powers unfolded [the *Φ*-structure]—an *intrinsic entity* that exists for itself, absolutely, rather than relative to an external observer” [12] (p. 28).

So, according to IIT, a PSC ontologically reduces to the unfolded structure of its specific, maximally integrated, intrinsic causal powers, i.e., its *Φ*-structure. At the same time, a *Φ*-structure ontologically reduces to its corresponding conscious experience (Section 3.3). Hence, ontologically speaking, there are only experiences; but from a third-person, physical perspective, experiences are operationalized in terms of cause–effect power as *Φ*-structures, which is how a physical substrate of consciousness (e.g., a conscious neural network) exists for itself, in physical terms [5,11,12,26]. In sum, while mereologically (maximally) irreducible (i.e., wholes that are maximally irreducible to their parts), PSCs ontologically reduce to their *Φ*-structures, which in turn, reduce to their corresponding consciousnesses. In other words, PSCs are not ontologically eliminated, they really exist. It is just that they are, intrinsically, truly, nothing but phenomenal entities, although described in operational, physical terms.

In synthesis, IIT’s *idealistic ontology* can be defined as follows:(i)**Phenomenal primitivism**: Conscious experiences are ontologically fundamental or primitive, neither deriving nor reducing their existence from/to anything else.(ii)**Eliminativism about non-PSCs**: Physical entities that do not specify maximal system integrated information (i.e., are not PSCs), such as, presumably, electrons, neurons, bodies, rocks and chairs, do not truly exist on their own (i.e., are ontologically eliminated; IIT’s “eliminative idealism” aspect).(iii)**Reductionism about *Φ*-structures**: Cause–effect structures do exist, but they are nothing but conscious experiences, albeit described in physical terms (i.e., *Φ*-structures are ontologically reduced to experiences; IIT’s “reductive idealism” about *Φ*-structures).(iv)**Reductionism about physical substrates of consciousness**: PSCs do exist, but they are nothing but *Φ*-structures seen extrinsically, and hence, are ultimately reduced to experiences also (i.e., IIT’s “reductive idealism” about PSCs).

It is important to highlight that the resulting metaphysical picture is wholly *monistic*, as claimed in [12,19,26]. In fact, it is a kind of radical monism according to which only subjective experiences truly exist as such; while the physical realm is either ontologically reduced to subjective experiences (*Φ*-structures and PSCs), or eliminated from existence (all non-conscious physical entities).

With all these distinctions in place, we turn now to the issue of how this radical idealistic ontology may be harmonized with IIT’s simultaneous endorsement of a kind of realism about the external world.

## 4. The Tension between IIT’s Idealistic Ontology and Its Realism

In the present section, we will examine in detail the important theoretical tension between IIT’s simultaneous endorsement of what we have called an idealistic ontology, and realism [11,12]. We begin presenting the tension in detail (Section 4.1), and then we reject some initially appealing potential solutions to dissolve the tension (Section 4.2). Finally, we offer what we think is the most straightforward and plausible solution; a non-solipsistic idealistic form of realism (Section 4.3).

### 4.1. The Tension Exposed

Intriguingly, although IIT has radical ontological implications regarding the primacy of experience and the secondary, either eliminable or reducible status of physical entities, the theory simultaneously endorses the thesis of *realism*, according to which

“We should assume that something exists (and persists) independently of our own experience… Although IIT starts from our own phenomenology, it aims to account for the many regularities of experience in a way that is fully consistent with realism” [12] (p. 6).

Prima facie, it seems very problematic to hold both that all that exist are ultimately conscious experiences, and that there are things existing independently of our consciousness. Recall that, according to IIT, “bodies and organs, tables and rocks…only exist vicariously… and so they do not truly exist” [11] (p. 8). How could the non-existence of all non-conscious entities be reconciled with the thesis of realism? More precisely, IIT is simultaneously claiming that (i) ultimately, only experiences exist (i.e., IIT’s idealistic ontology), and that (ii) there is an experience-independent existence (i.e., IIT’s realism). However, (i) and (ii) seem to contradict each other. If only experiences ultimately exist, then there is no experience-independent existence. Conversely, if there are things existing independently of any experience, then it could not be the case that ultimately, only experiences exist. In sum, if (i) is true then (ii) is false; or, if (ii) is true, then (i) is false.

### 4.2. Potential Replies That Do Not Work

A first potential reply could be that, in contrast with IIT’s idealistic ontology, realism is not really an ontological claim, but merely one of its “methodological guidelines” [12] (p. 6). According to this line of thought, the thesis of realism is just a working assumption, a good inference to account for and pragmatically deal with the many regularities of experience in relation to what seems to be an independently existing external world. In other words, it would just be a heuristic principle aiding in practical matters. This is consistent with IIT’s rejection of solipsism (i.e., only my consciousness exists), which is based on explanatory rather than ontological grounds: “[realism] is a much better hypothesis than solipsism, which explains nothing and predicts nothing” [12] (p. 6). Nonetheless, regarding ontology, IIT’s realism may be neutral.

Our objection is that even if realism is meant to be a “methodological guideline”, it would be a metaphysical thesis nonetheless: it claims that we should believe in the existence of an external world beyond our minds. If this is not a paradigmatic example of a metaphysical belief, then we do not know what else could be. So, it can be granted that IIT’s realism could be a heuristic, fundamentally aimed at pragmatically guiding the scientific study of consciousness. However, this qualification would merely affect the *attitude* of the researcher towards the thesis of realism (i.e., “act as if you believe it”), but not the *content* of the thesis, which would remain metaphysical all along. Hence, at the metaphysical level of discourse, IIT’s idealistic ontology and realism would remain in a theoretical discrepancy. By entailing that only experiences truly exist, IIT’s account seems fully inconsistent with the metaphysical claim that an external, physical world exists experience-independently.

A second potential reply from IIT theorists may be the following. Non-conscious entities such as bodies and chairs do exist, it is just that they exist extrinsically, not for themselves but from the perspective of an intrinsic entity. In other words, extrinsic existence would not be non-existence. Although we see here a promising route to resolve the problem (see [40]), as stated, it does not work. Realism comprises not only the belief in the *existence* of the things and properties of a given domain one is realist about, but also the belief in their *mind-independence* [30], which is not the case for extrinsic entities in IIT. It would be similar to claiming that unicorns do exist, but just not mind-independently. Even if that could be granted, it would be insufficient to ground realism about unicorns. According to IIT, systems that do not specify maximal *φ*_s_ (i.e., every non-conscious entity), at best, “only exist relatively, for an observer” [11] (p. 8), that is, mind-dependently, which is contrary to what realism requires.

A third potential reply is Kantian in flavour. It says that causally powerful extrinsic entities do exist mind-independently, as things-in-themselves or *noumena*, but we will never have direct cognitive or experiential access to them as such. At best, we know them as they appear to us, i.e., as *phenomena*. A first difficulty with this reply is that it would entail that we cannot have substantive knowledge about the metaphysical nature of extrinsic entities, because the way they are-in-themselves would be forever beyond our grasp, and hence, we would need to remain silent about their true nature. However, IIT does not remain silent about this at all, claiming instead that the divide between intrinsic and extrinsic entities is a divide of being, not of knowing. In other words, there is an *epistemic contradiction* between the Kantian suspension of judgment about the true noumenal nature of extrinsic entities and IIT’s purported knowledge about it. This is closely related to our second worry. If extrinsic entities do not truly exist, for themselves, but only relative to conscious observers, then intuitively, they would not exist-in-themselves either, as noumena beyond how they appear to us, i.e., they would be purely phenomena. In other words, there is a *metaphysical contradiction* between the Kantian reply that extrinsic entities have a noumenal, albeit, unknowable nature, and IIT’s implication that they have no noumenal nature at all.

### 4.3. The Straightforward Solution: Non-Solipsistic Idealist Realism

We claim that there is at least one straightforward plausible alternative for IIT to deal with this apparent contradiction. The solution is to take other experiences as the only type of entities that exist independently of one’s own experience. In other words, our proposal interprets IIT’s realist statement that “something exists independently of our own experience” [12] (p. 6) as simply claiming that “something” means here “other experiences”, which exist independently of one’s own. Note that this interpretation advocates a form of realism that diverges sharply from traditional forms of realism according to which the external world is fundamentally physical and mind-independent. In contrast, our reading of IIT’s realism claims that the external world does exist independently of one’s own consciousness, but is fundamentally constituted by other experiences beyond one’s own.

To clarify our interpretation of IIT’s realism as the most plausible solution to the tension we are dealing with, and its plausibility as a form of realism, it is useful to evaluate it in light of Chalmers’ distinction between *realist* and *anti-realist* versions of idealism [15]. According to the philosopher, in the anti-realist idealistic picture, “there is no concrete reality external to how things appear: all concrete non-mental truths p are grounded in or constituted by appearances that p, or in closely related truths involving appearances” [15] (p. 592). In other words, reality is exhausted by how things appear to conscious minds, with no additional nature beyond those appearances. A paradigmatic example would be Berkeley’s subjective idealism, captured by the slogan *esse est percipi* (to be is to be perceived), according to which things exist only insofar they are perceived by a mind.

In the realist idealism, in contrast, the claim that reality is exhausted by how things appear to conscious minds is rejected; “the physical world really exists out there, independently of our observations; it just has a surprising nature” [15] (p. 592). This “surprising nature” is, of course, mental. Examples of this view are panpsychist variants of idealism in which the fundamental constituents of the physical world are conscious microsubjects; or cosmopsychist idealisms, in which the whole of physical reality is grounded in a cosmic consciousness [41,42,43,44,45]. According to Chalmers, these idealist views are realist, because physical reality is not constituted by how things appear to a conscious mind, but “it is the structure and relations among experiences rather than their specific content that matters [to ground physical reality]” [15] (p. 592).

With this distinction in place, we believe that, according to our proposed understanding of IIT 4.0′s ontology (Section 3) and corresponding version of realism (this section), IIT’s metaphysics can be conformingly categorized as a *realist idealism*, with respect to Chalmers’ classification. The nature of conscious (subregions of) brains, conscious neuromorphic hardware, or even minimally conscious photodiodes, according to our proposed reading of IIT, is certainly not exhausted by how they appear to external observers. That is, their nature is not even closely conveyed by being, say, a wrinkled, pinkish-grey mass (brain), a complex network of tiny, interconnected metallic components (neuromorphic hardware), or a small, shiny rectangle, encased in a protective material (photodiode). Nor would their true nature be expressed by a more refined and technical vocabulary along the same lines. Their true nature is how they exist for themselves, that is, what it is like to be them from their own perspective.

So, in line with Chalmers’ account of realist idealism, these physical substrates of consciousness (excluding my own PSC), truly exist independently of my own subjective consciousness and not as how they appear to me. It is just that their nature is eminently experiential, they are definite, specific, and irreducible structures of intrinsic phenomenal existence. Thus, while the PSC of my own consciousness is just a way to interact and describe my consciousness from the outside, the PSCs of other systems’ experiences are ontologically independent of my own consciousness, they really exist out there in the world, it is just that their true nature is not physical but experiential.

This interpretation respects IIT’s rejection of solipsism, which seems to be a core motivation for endorsing realism in the first place [12] (p. 6). According to this reading, not only does my consciousness exist, or better, Giulio Tononi’s consciousness (i.e., solipsism is rejected), but also other consciousnesses: the subjective experiences of every conscious living human being, animals, other living organisms and even simple artificial systems such as photodiodes [27,46]. In other words, the “external world” really exists beyond my own subjectivity, it is just that it is fundamentally constituted by other subjective experiences, which can also be referred to, more theoretically, as “*Φ*-structures”, or as “physical substrates of consciousness” (PSCs).

In this sense, the more traditional form of realism is discarded, namely, the view according to which the external world really exists independently of one’s own mind and is fundamentally constituted by non-conscious physical entities such as quarks and electrons. In accordance with IIT’s idealistic ontology, especially due to its principle of true existence (Section 3.1), quarks, electrons, neurons, bodies, and stars do not exist as mind-independent entities, but only as extrinsic entities that specify, at best, non-maximal *φ*_s_ values, and that, hence, exist only relative to conscious observers (mind-dependently). As such, our idealistic reading of IIT’s realism is the only one that would respect the theory’s “great divide of being” [11] (p. 8) between truly existing, intrinsic, conscious entities, and untruly existing, extrinsic, non-conscious entities that could only exist relative to some consciousness.

In sum, we regard our idealistic interpretation of IIT’s realism as the most parsimonious solution to the logical tension we exposed in Section 4.1 (between the theory’s idealistic ontology and realism) because it allows IIT to simultaneously endorse, without contradiction, its idealistic ontology, and a compatible, non-solipsistic version of realism, without requiring any amendment to the theory’s core assumptions.

## 5. Concluding Remarks and Future Work

In the present article, we have tried to advance a better philosophical understanding of IIT 4.0, centred on the apparent tension between its idealistic ontology and realism. After presenting the fundamentals of the theory (Section 2), we argued that IIT’s idealistic ontology should be understood as a combination of phenomenal primitivism, reductionism regarding *Φ*-structures and physical substrates of consciousness (PSCs), and eliminativism about non-conscious physical entities (Section 3). That is, the theory’s ontology asserts that only experiences ultimately exist, but that these can also be described and purportedly explained scientifically as *Φ*-structures, and more indirectly, as physical substrates of consciousness. Identifying and understanding the latter would be the typical target of standard neuroscientific research on consciousness, e.g., finding the brain network minimally sufficient to support consciousness. However, according to IIT, the way a neural substrate exists for itself, as a subjective consciousness, is far better conveyed scientifically by the theory’s notion of *Φ*-structure and its causal properties. In other words, a PSC truly exists, intrinsically, as a *Φ*-structure, which means as a conscious entity with such and such phenomenal structure. Thus, the terms “conscious experience” and “*Φ*-structure” are two ways to describe how a conscious entity exists for itself: phenomenologically in the case of the former; physically/scientifically in the case of the latter. On the other hand, IIT’s eliminativist aspect asserts that all systems that do not specify maximal system integrated information do not exist as conscious, intrinsic entities, and hence, do not truly exist as entities on their own (mind-independently). The entities that are ontologically eliminated include, presumably, atoms, fMRI scanners, cerebellums, living bodies, and distant galaxies, among many others (i.e., all non-PSCs).

Then, in Section 4 we highlighted the tension that this metaphysical position entails regarding IIT’s own declared realism. After presenting and refuting three potential solutions to this apparent contradiction, we proposed what we regard as the most plausible alternative: understanding IIT’s realism as an assertion of the existence of other experiences beyond one’s own, what we have called a *non-solipsistic idealist realism*. We also evaluated the resulting metaphysical picture in light of Chalmers’ [15] distinction between realist and anti-realist idealisms and concluded that it qualifies as a species of the former. In other words, according to our interpretation of IIT’s mathematically formalized metaphysics, the conscious brain regions of other beings beyond one’s self truly exist out there in the world, independently of one’s own consciousness, and not as how they appear to oneself. It is just that their true nature is experiential, they are intrinsically existing phenomenal structures.

In sum, we claim that the overall metaphysical picture we have proposed clarifies and further advances our understanding of IIT’s ontological implications; and moreover, it resolves the fundamental tension between its idealistic ontology and realism, showing the kind of realism that is compatible with IIT’s ontology. Nevertheless, future work should address several other issues that arise from IIT’s ontological elimination of every non-conscious entity [40]. For instance, IIT’s metaphysics *prima facie* imply that an engineer can create a conscious physical system (a system with maximal *φ*_s_) using individual components that did not exist (physical stuff that did not specify maximal *φ*_s_), such as separate logic gates and wires. In other words, it seems that existence can be engineered from non-existence; a very uncomfortable implication. Additionally, the common belief that consciousness originated at some point in the evolution of life, under IIT’s metaphysics, apparently means that consciousness originated from nothing (i.e., non-truly existent non-conscious life forms); again, a very unpalatable implication. Importantly, IIT’s notion of an “ontological dust” [11] may be useful to handle these issues. According to this, separate logic gates and wires, as well as non-conscious living beings, may not be completely non-existent after all, because it may be theoretically possible that they are aggregates of intrinsically existing atomic constituents, such as minimally conscious elementary particles, fields or molecules.

In future work [40], we will address these and other related problems in IIT’s realist idealism (as defined here), as well as ways in which IIT may overcome them, including the “ontological dust” alternative. However, we are initially inclined to think that the theory may better be revised in some specific conceptual/metaphysical aspects in order to avoid the elimination of all non-conscious physical entities and instead embrace a corresponding expanded form of realism. In addition to other experiences beyond one’s own, this expanded realism would assert the mind-independent existence of non-sentient organisms, pieces of wire, neurons and fMRI scanners, among other non-conscious physical entities, as long as they are causally powerful. Arguably, this may result in a better version of IIT, one which would be more aligned to an ontological form of emergentism [6,47]. This emergentist IIT could provide, given the theory’s robust causal formalism [21,48,49,50,51], a crucial aid to advance our scientific understanding of the causal difference that consciousness seems to make in a diversity of phenomena such as motivation [52,53,54,55,56], psychedelic medicine [57,58,59,60], information integration and behavioural flexibility [61], free will [62,63,64,65], and other psychological and behavioural functions [66,67,68,69], thus holding the potential to significantly advance our scientific knowledge of consciousness and its place in nature.

## Data Availability

Not applicable.

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
