# Peer review of "The Fundamental Tension in Integrated Information Theory 4.0’s Realist Idealism"

_entropy, 2023, doi:10.3390/e25101453_

Round 1

Reviewer 1 Report

The paper develops a solution for a philosophical problem regarding IIT (integrated information theory), which at first sight depends on inconsistent assumptions: nothing exceptional personal (conscious) experience exists, and there is an independent reality. The proposal is to claim that an individual recognises the consciousness of other agents, be these human or not. 

The explanation of the problem and its context are clear and convincing. When it comes to the solution, my impression is that the authors view their portal with great suspicion. In the concluding section a suggestion is made to adopt some form of realism anyhow. I think the paper can be improved by explaining better how this second discrepancy must be handled: the main result of the paper is al but discredited in the final section.

I must admit that I have difficulties in imagining a demarcation of which agents or entities are supposed to be able to create forms of existence through having experiences and being conscious.  Are all bacteria included in this category of agents, there ought to be some kind of limit.

The paper can be improved by being more clearcut about the value of the proposed solution in the light of the fact that theory development in the eyes of the authors (as far as I understand them well) should move in another direction.

I suggest to add some lines to the paper regarding these questions, which I assume to constitute a minor revision.

Author Response

We thank Reviewer #1 for his/her kind words and helpful comments.

Here we address the reviewer’s main point that “The paper can be improved by being more clearcut about the value of the proposed solution in the light of the fact that theory development in the eyes of the authors (as far as I understand them well) should move in another direction”

In the revised manuscript, concluding section, we are now clearer about what we accomplished in the present paper, namely, the philosophical clarification of IIT 4.0’s ontological implications and the form of realism it is compatible with. At the same time, we are now more precise about distinguishing this and the further philosophical work that is still needed, which we are going to address in a future study.

More specifically, we added to the concluding section the following:

In sum, we claim that the overall metaphysical picture we have proposed clarifies and further advances our understanding of IIT’s ontological implications; and moreover, it resolves the fundamental tension between its idealistic ontology and realism, showing the kind of realism that is compatible with IIT’s ontology. Nevertheless, future work should address several other issues that arise from IIT’s ontological elimination of every non-conscious entity [40]. (Lines 660-5)

Also, we changed the name of the concluding section from “Concluding remarks” to “Concluding remarks and future work” (L. 622), to further clarify that we are also introducing some additional issues that the present work didn’t address and that we analyze in future work.   

Many thanks again to Reviewer #1 for pressing us to clarify this.

Reviewer 2 Report

This philosophical paper aims to elucidate the ontological implications of IIT. To that end, the authors analyzed the existing IIT literature and discussed it in light of various philosophical categories that could potentially be applied.  

As a long-time IIT contributor, I enjoyed the exposition and largely agree with the authors' conclusions. I did not find any flaws in their presentation of IIT. 

I have one comment that the authors may wish to address:

It seems clear that IIT accepts the mind-independent existence of other experiences than my own. It's also clear that IIT's ontology is sparse: the constituents of a substrate of consciousness cannot be conscious themselves, so they do not exist (intrinsically). Neither would an aggregate of conscious entities exist (intrinsically). However, while the quarks, electrons, neurons etc in my PSC are excluded from existing, those neurons that are not part of a PSCs could form their own maxima of phi_s.

What I am getting at is that the iterative algorithm to identify conscious entities (see 4.0) might in the end cover (literally) everything in such a way that every physical micro unit is either a constituent of a conscious entity or a conscious entity in itself (ontological “dust”, see Free Will paper p. 7). This depends on the nature of the micro TPM of our universe. It would require micro units to have a little bit of noise, but not be completely random. (Whether physics allows for such a thing is another question.) Such a picture would alleviate the “something out of nothing” worry (l. 658-665). I think this is worth mentioning as an option.

A “non-solipsistic idealist realism” that leaves nothing unaccounted for would be the most coherent picture in my opinion.

Alternatively, I still think that it is a viable option (compatible with IIT) to grant a secondary kind of existence to the “(micro) stuff out there”, which is not the same as the notion of “extrinsic existence”. Maybe the authors follow up paper [40] will be a proposal along those lines.

There are a few typos, e.g. l. 216 "amounts to" --> "amount to"

Author Response

We deeply thank Reviewer #2 for his/her helpful comments and insightful suggestions, which we believe helped improve the paper.  

Here we address the reviewer’s main point that “the iterative algorithm to identify conscious entities (see 4.0) might in the end cover (literally) everything in such a way that every physical micro unit is either a constituent of a conscious entity or a conscious entity in itself (ontological “dust”, see Free Will paper p. 7)… Such a picture would alleviate the “something out of nothing” worry (l. 658-665). I think this is worth mentioning as an option.”

In the revised manuscript, in the concluding section, we now explicitly mention and explain IIT’s notion of the “ontological dust”, and how this may be helpful to address the issues we mentioned that were still remaining and that future work will address. Also, we made several changes of emphasis in the same section, to suggest that IIT may have the resources to overcome these problems (such as the “ontological dust” notion), but that nonetheless further work is required to assess this. Also, we acknowledge that although IIT might be able to handle these apparent issues, we are initially inclined to think that the theory needs some conceptual-metaphysical changes to expand the scope of its realism to include non-conscious (not even minimally-conscious) physical entities with causal power.

More specifically, we added to the concluding section the following:

Importantly, IIT’s notion of an “ontological dust” [11] may be useful to handle these issues. According to it, separate logic gates and wires, as well as non-conscious living beings, may not be completely non-existent after all, because it may be theoretically possible that they are aggregates of intrinsically existing atomic constituents, such as minimally conscious elementary particles, fields or molecules.      

In future work [40], we will address these and other related problems in IIT’s realist idealism (as defined here), as well as ways in which IIT may overcome them, including the “ontological dust” alternative. However, we are initially inclined to think that the theory may better be revised in some specific conceptual/metaphysical aspects in order to avoid the elimination of all non-conscious physical entities and instead embrace a corresponding expanded form of realism. (Lines 672-684)

Regarding the minor improvements of English language, we also made a line-by-line review and revised several minor mistakes throughout the text, such that the language quality now seems to be flawless.  

Many thanks again to Reviewer #2 for his/her commentaries and suggestions. 

Round 2

Reviewer 2 Report

The authors have addressed my comments in full.